# DPO: Direct Planar Odometry with Stereo Camera

**DOI:** 10.3390/s23031393

**Published:** 2023-01-26

**Authors:** Filipe C. A. Lins, Nicolas S. Rosa, Valdir Grassi, Adelardo A. D. Medeiros, Pablo J. Alsina

**Affiliations:** 1Federal Institute of Rio Grande do Norte, Parnamirim 59143-455, Brazil; 2Department of Electrical and Computer Engineering, University of São Paulo, São Carlos 13566-590, Brazil; 3Department of Computer Engineering and Automation, Federal University of Rio Grande do Norte, Natal 59078-970, Brazil

**Keywords:** visual odometry, direct method, planar features, second-order optimization, stereo camera

## Abstract

Nowadays, state-of-the-art direct visual odometry (VO) methods essentially rely on points to estimate the pose of the camera and reconstruct the environment. Direct Sparse Odometry (DSO) became the standard technique and many approaches have been developed from it. However, only recently, two monocular plane-based DSOs have been presented. The first one uses a learning-based plane estimator to generate coarse planes as input for optimization. When these coarse estimates are too far from the minimum, the optimization may fail. Thus, the entire system result is dependent on the quality of the plane predictions and restricted to the training data domain. The second one only detects planes in vertical and horizontal orientation as being more adequate to structured environments. To the best of our knowledge, we propose the first Stereo Plane-based VO inspired by the DSO framework. Differing from the above-mentioned methods, our approach purely uses planes as features in the sliding window optimization and uses a dual quaternion as pose parameterization. The conducted experiments showed that our method presents a similar performance to Stereo DSO, a point-based approach.

## 1. Introduction

A requirement for a mobile robot’s autonomy is the ability to extract useful information from the environment and create a map by using its sensors, while the robot explores the environment and simultaneously locates itself with that map. The research areas that address this problem using cameras as the main sensor are called Visual Odometry (VO) and visual Simultaneous Localization and Mapping (vSLAM).

According to Chen et al. [1], these techniques can be classified into two main different approaches: feature-based or direct techniques. The first is based on the extraction of features present in successive images to, through correspondence between these features, determine the pose of the robot and reconstruct a sparse map. The second approach minimizes the photometric error between a set of pixels of two images using a homography function. Both techniques are used in either monocular, stereo or RGB-D camera systems.

While early works paid more attention to the quality of pose estimates, with the extraction of sparse environment information, recent works have focused on improving the quality and density of the mapping in the SLAM process. More accurate sensors, computer processing power increase and new techniques contributed to the emergence of (semi-)dense and/or large-scale maps in VO and visual SLAM systems [2]. Now, some techniques allow choosing the density of the map according to the computational resources available [3].

These maps can be constructed with many types of features present in the environment, starting from geometric primitives, such as points [4], lines [5] and planes [6], to more complex structures, for instance, quadrics [7], cubics [8] and objects detected via Convolutional Neural Networks (CNNs) [9].

A predominance of techniques based on keyframes is observed in direct methods [10,11], which in general store point information in certain frames along a path traveled by the robot. Thus, point-based maps are widely used in such techniques. One of the reasons for this is the fact that high-gradient regions are present in corners and edges, which can be used to obtain disparity maps and create point-based maps. However, the use of points for map representation could be computationally expensive as far as the number of points increases and they do not contribute to the overall interpretation of the 3D structure of the mapped environment since each point is represented independently.

A more appropriate map representation, when the desired requirement is to use features that optimize the relationship between density and stored information, is to make use of planar regions. Planes are dominant in structured environments, both internal and external, and can be expandable, representing a large part of the scene with a few parameters [7]. Planes make possible the use of some constraints, such as Manhattan constraints [6,7], to restrict the relationship between planes in the environment, producing more consistent maps. Furthermore, it is possible to assign semantics to planar regions, since a region can represent, for example, walls or the floor and a set of planar regions can represent more complex structures [12].

The most prominent direct VO approach in recent years was developed by Engel et al. [10], called Direct Sparse Odometry (DSO). This technique initially developed for monocular cameras is based on point cloud and has become popular due to the quality of the results obtained in estimating odometry with reduced information. Several works have been developed from this technique. However, only recently, works developed by Wu and Beltrame [13] and Xu et al. [14] presented the first methods that use planes within the DSO framework.

In the first mentioned planar technique, the inverse depth of a pixel, used as the feature parameter in the DSO method, is expressed as a function of the homogeneous plane variables. This plane representation can also be understood as the coefficients of a unit quaternion [15]. Although this technique presents a valid description of the use of planes in the DSO framework, it requires that sufficiently good initial estimates of planes be obtained for the optimization to converge. To solve this problem, this technique made use of a plane estimator based on a CNN, which restricts the whole approach to the data domain used to train the network. In addition, it causes the method to depend on the quality of network predictions to initialize properly.

The second mentioned planar technique uses points that are part of the same plane to add a constraint to the optimization and increase the accuracy of the odometry results. This technique prioritizes indoor environments as far as it detects vertical and horizontal planes.

A representation of planes that is also minimal and follows the natural evolution of the DSO approach is the inverse depth of three image points [16]. The use of this approach makes possible the employment of disparity-based algorithms, which estimate point depth, already used by DSO. Furthermore, this approach causes the map features to be interpreted in two ways, i.e., as a set of three points or as a planar region formed by three vertices. This dual interpretation can be advantageous in situations in which point segmentation algorithms in image planes fail, causing points from different planes to be present in the same segment.

Thus, inspired by the DSO approach, the goal of this paper is to present a purely direct stereo VO framework that performs mapping with planar regions, represented by three-point inverse depths.

The proposed framework is, to the best of our knowledge, the first stereo plane-based VO inspired by the DSO framework. Moreover, we first introduce the use of dual quaternions in parameterizing the robot pose for direct methods. According to Wang et al. [17], among the various existing rigid body representations, the unit dual quaternion offers the most compact and computationally efficient screw transformation formalism.

This paper is organized as follows. Section 2 is a literature review of the main works in the area of direct VO and visual SLAM. In Section 3, a brief introduction of the main theoretical components addressed in this paper is presented. Then, Section 4 presents the Direct Planar Odometry framework. Section 5 presents the results of the proposed technique, as well as a discussion about them. Finally, in Section 6, we highlight the main ideas and contributions of this work and present some prospects for future works.

## 2. Related Works

This section presents a literature review of the developed techniques of Direct VO and visual SLAM. These techniques are classified according to the generated map as point-based or plane-based. Subsequently, SLAM methods that use dual quaternions for pose parameterization are presented. Finally, the main contributions of this paper are listed.

### 2.1. Point-Based Direct VO/SLAM

One of the most popular approaches has been the technique developed for monocular cameras called Large-Scale Direct SLAM (LSD-SLAM) [18]. LSD-SLAM presents a direct SLAM framework that generates a consistent large-scale map. For this, a semi-dense depth map, obtained from high-gradient pixels, is used in the image alignment. In the pose parameterization, the Lie group SIM(3) is applied and in addition to the rigid body parameters (rotation and translation), a scale parameter is added. Finally, this method achieves a consistent global map formed by 3D points through pose-graph optimization. A stereo camera version [11] and an extension of it [19], as well as an omnidirectional camera approach [20], have also been developed.

Recognizing some deficiencies in LSD-SLAM, mainly in pose estimation, some researchers have proposed improvements to the framework. Wang et al. [21] proposed to improve the initial estimation of VO by incorporating a constant-velocity model to provide initial estimates for the optimization, whereas Zhang et al. [22] developed a learning-based confidence model, which applied to the depth map helps in choosing the best pixels to be used in the optimization.

Later, the same research group that developed LSD-SLAM proposed DSO, which brought significant improvements in monocular odometry by optimizing camera poses, intrinsic and map parameters, boosting results through the use of photometric camera calibration. Subsequently, several versions of this work were presented, mainly addressing the use of different camera types [3,23], a modification with loop closure [24] and using predictions provided by deep learning techniques [25]. Yang et al. [26] proposed a semi-supervised network called StackNet for deep monocular depth estimation; these estimates were incorporated to overcome scale drift, an intrinsic limitation in monocular VO. The so-called Deep Virtual Stereo Odometry (DVSO) framework incorporates the left disparity prediction in the depth initialization and the right prediction in the bundle adjustment process using a virtual stereo term. More recently, Yang et al. [27] developed the D3VO, which integrated three pieces of information predicted by a semi-supervised network to benefit a monocular odometry method. More specifically, they used a network based on MonoDepth2 [28] which, besides predicting depth and relative camera pose, also retrieves photometric uncertainty and brightness transformation parameters. These predictions were then used through the processing framework, respectively, serving as initial values for both the front-end tracking and back-end optimization, performing inter-frame illumination alignment and weighting the photometric residuals.

Other approaches based on the DSO technique have also been developed. One of these approaches was SalientDSO [29] which, inspired by human vision, searches for regions of interest in the image that serves as input to the system. Parallelization of the original approach was proposed by Pereira et al. [30]. In addition, semi-direct approaches that mix direct VO methods and indirect ones to refine keyframe pose are also present in the literature [31,32,33].

### 2.2. Plane-Based Direct SLAM

One of the first real-time direct SLAM techniques to gain notoriety was the work proposed by Silveira et al. [16], who developed a monocular SLAM, using planes to generate the map, optimized by Efficient Second-order Minimization (ESM). To obtain initial estimates of the parameters, an approach that couples the proposed direct technique with a filter-based method was adopted.

Plane parameterization using the depths of three points, represented by three non-collinear pixels in a given image patch, is also introduced in the mentioned approach. This representation was employed on several square-shaped patches, subdividing the image, to estimate the plane parameters. In some situations in which patches are projections of regions with more than one plane in the scene, not necessarily all pixels of the patch will be contained in the same plane as the three points used in the parameterization. Such pixels can affect the correct estimation of the parameters.

In the approach DPPTAM [34], a monocular technique based on LSD-SLAM is developed using a semi-dense depth map. In the mentioned paper, the image is segmented using superpixels [35] and the points of the semi-dense map are projected onto the image and clustered according to plane parameterization segmentation performed by the superpixels. After that, the planes are estimated using Singular Value Decomposition (SVD) applied to the points clustered in each superpixel. DPPTAM keeps the points of a superpixel, allowing the use of non-convex regions. However, this is not a compact representation, since all the points of the superpixel are stored. In other words, the same problem of point-based visual SLAM techniques remains.

Ma et al. [36] developed CPA-SLAM, a real-time visual SLAM technique developed for RGB-D cameras that mix photometric and geometric alignment in the same optimization. To estimate planes, the keyframe is segmented into many regions and a global plane model that uses graph optimization is applied in all local observations.

As mentioned previously, the recent work by Wu and Beltrame [13] presented the first planar DSO-based technique. For each point chosen at each new image, parameters of the plane represented by unit quaternions are estimated. When a set of neighboring points projected from different keyframes in the current frame have the same descriptor, a planar region is determined and inserted into the optimization. The prediction strategy is performed using the PlaneNet [37] technique, which estimates the planes present in each pixel from a color image. This CNN-based technique is time-consuming and its efficiency depends on the domain of the network training data, which in practice limits the applicability of the framework as a whole.

The second planar DSO-based approach (PVI-DSO) is presented by Xu et al. [14], who developed a visual-inertial odometry (VIO) technique that fuses coplanar constraint regularities to improve the accuracy of the localization. In this application, only vertical and horizontal planes are detected, prioritizing its use in structured environments.

All approaches mentioned above showed a preference for point-based maps in direct SLAM applications. Among the few direct techniques found that use planes, none of them is a stereo-based framework.

### 2.3. Pose Parameterization

Different parameterizations for the pose have been used in visual SLAM and VO techniques. The homogeneous transformation matrix T∈SE(3) alongside its algebra se(3) has been one of the preferred approaches of the main direct SLAM methods. Virtually all of the techniques presented in this paper use this approach or a similar one, such as SIM(3), which takes into account the scale factor.

One parameterization that is not new but that has recently been employed in visual SLAM techniques is *unit dual quaternion*. According to Thomas [38], quaternions are of vital importance in representing spatial rotations; however, a lack of precise understanding of the meaning of its operations leads to an underutilization of quaternion-derived techniques, such as *dual quaternions*.

According to Kenwright [39], dual quaternions are compact, unambiguous, singularity-free and computationally minimalist for rigid transformations. In addition, they offer the most compact way to represent rotations and translations. More recently, they have been employed in visual SLAM techniques based on stochastic filters [40,41,42,43,44,45], and some based on graph-SLAM [46,47] have been observed. Despite the wide use of this parametrization in feature-based SLAM approaches, our approach is the first to use it in direct VO.

### 2.4. Contributions

The main contribution of this work is the development of a plane-based direct VO technique inspired by the DSO approach. To the best of our knowledge, our work is the first purely direct stereo VO approach to use planes as features. The proposed plane parameterization represents a natural evolution of the DSO technique and has the following qualities:From the pose optimization point of view, the vertices of the planar region do not need to be part of the same real scene plane. For these cases, it is possible to interpret the feature as a set of three points. Thus, there is no need for differentiation of points and planes in the mathematical formulation of the problem.As a consequence of the previous statement, this parameterization allows the use of fast but low accuracy, algorithms for segmenting pixels into planes, not hindering the performance of the technique as a whole.Inspired by the DSO technique, which uses only a set of 8 pixels per point in the parameter optimization, this work uses only 8 pixels per vertex of the planar region, which represents only 24 pixels per plane. Depending on the distance between vertices, it can describe a large planar area in the scene with a small amount of data.This approach allows the three vertices of an existing planar region to be reused to form new planar regions, even contributing to a better coupling between the map elements.

A second contribution is the use of dual quaternion for representing the pose, that being the first direct VO technique that uses this type of parameterization.

## 3. Theoretical Background

In this section, the basic concepts for the development of this work are presented, starting with the homography model, which is dependent on the camera pose and the planes in the environment. Then, the basic concepts of the unit dual quaternion, the plane parameterization and the efficient second-order optimization model are presented. Some of these topics have been presented in a previous study by the authors, which compares different parameterizations of a plane [48]. However, for didactic purposes, we decided to present these concepts again in this article.

### 3.1. Plane-Based Two-View Homography

Figure 1 shows the projections of point $P={\left[x,y,z,1\right]}^{⊤}$, belonging to the plane π, on the reference image $I^{*}$ and the image I. It is assumed that the projections $p^{*}={\left[u^{*},v^{*},1\right]}^{⊤}\propto K\left[I_{3}\;0\right]P$ and $p={\left[u,v,1\right]}^{⊤}\propto K\left[R\;t\right]P$, where $I_{3}$ is a 3×3 identity matrix, K is the intrinsic parameters matrix, R is the rotation matrix and t is the translation vector. The equation that maps $p^{*}$ to p is given by:(1)$$p\propto KRK^{-1}p^{*}+z^{*-1}Kt,$$
where $z^{*-1}$ is the inverse depth of the point P, relative to the reference frame $I^{*}$ and, in this case, ∝ means an up-to-scale equality.

The relationship between the plane π and the inverse depth $z^{*-1}$ is given by:(2)$$z^{*-1}=\frac{1}{\rho^{*}}n^{*⊤}K^{-1}p^{*},$$
where plane π is parameterized by the unit normal vector $n^{*}$ and its distance $\rho^{*}$ from the reference frame. Thus, substituting $z^{*-1}$ in Equation (Equation 1) by Equation (Equation 2), we find:(3)$$\begin{array}{cc} H & =K\left(R+t\cdot n_{\rho }^{*⊤}\right)K^{-1},\\ p & \propto Hp^{*}, \end{array}$$
such that $n_{\rho }^{*}=n^{*}/\rho^{*}$ and H is the homography matrix. Finally, multiplication between H and $p^{*}$ leads to:(4)$$p=w\left(H,p^{*}\right)={\left[\begin{array}{ccc} \frac{h_{11}u^{*}+h_{12}v^{*}+h_{13}}{h_{31}u^{*}+h_{32}v^{*}+h_{33}}; & \frac{h_{21}u^{*}+h_{22}v^{*}+h_{23}}{h_{31}u^{*}+h_{32}v^{*}+h_{33}}; & 1 \end{array}\right]}^{⊤},$$
where $h_{ij}$ means homography matrix element.

The parameters of the homography matrix H are $n_{\rho }$, representing plane information, R and t, representing a pose. Since the front-end of this work estimates pose and plane parameters separately, the optimization formulation of the plane and pose will be presented independently.

For plane estimation, the pose between two images is considered to be known. This relative pose can be static, i.e., given by the stereo baseline, and temporal, given by successive images of the left camera. Thus, Equation (Equation 3) becomes:(5)$$p\propto H\left(n_{\rho }^{*}\right)p^{*}.$$

In the step for obtaining the relative pose of subsequent images from the left camera, the structure information is considered known and Equation (Equation 3) becomes:(6)$$p\propto H\left(R,t\right)p^{*}.$$

### 3.2. Problem Formulation

In contrast to the DSO approach, the optimization technique used to estimate planes and poses in this work is Efficient Second-order Minimization (ESM). This second-order technique, presented by Malis [49], has as its main advantage the need to determine only first-order approximations.

The photometric error is formulated as a non-linear optimization problem and it is defined as:(7)$$\min_{\zeta ,\;\theta }=\frac{1}{2}\sum_{i=0}^{n-1}{\left\{I\left[w(H\left(T\left(\zeta \right),n_{\rho }^{*}\left(\theta \right)\right),p_{i}^{*})\right]-I^{*}\left[p_{i}^{*}\right]\right\}}^{2},$$
where ζ represents pose parameters and θ represents plane parameters.

#### 3.2.1. Plane Optimization

The plane is estimated by the optimization considering the relative pose between a given frame and a reference frame is known. Therefore, Equation (Equation 7) is rewritten to include only $n_{\rho }^{*}$ as parameter in the optimization:(8)$$\min_{\theta }=\frac{1}{2}\sum_{i=0}^{n-1}{\left\{I\left[w(H\left(n_{\rho }^{*}\left(\theta \right)\right),p_{i}^{*})\right]-I^{*}\left[p_{i}^{*}\right]\right\}}^{2},$$
where the photometric error element $d_{i}\left(\theta \right)=I\left[w\left(H\left(n_{\rho }^{*}\left(\theta \right)\right)\right)p_{i}^{*})\right]-I^{*}\left[p_{i}^{*}\right]$ is part of the residual vector d(θ) of size (n×1) and θ represents the vector of parameters of $n_{\rho^{*}}$, such that:(9)$$n_{\rho }^{*}\left(\theta \right)=g\left(\theta \right),$$
where g(·) is the function that maps the parameter vector of the plane θ to its normal representation $n_{\rho }^{*}$, applied in Equation (Equation 3).

Similar to other optimization methods (e.g., Newton), ESM is an iterative method. To determine the new estimate $\theta_{k+1}$, a function dependent on the previous estimate $\theta_{k}$ and the Δθ increment is applied:(10)$$\theta_{k+1}\leftarrow f\left(\theta_{k},\Delta \theta \right),$$
where f(·) represents the theta update function, which depends on the relation between $n_{\rho }^{*}$ and θ. The optimization will estimate new values of θ until the established stopping criterion is satisfied.

On the assumption that d(θ) is expanded in a second-order Taylor series around the point $\theta =\hat{\theta }$, we have:(11)$$\begin{array}{cc} d\left(\theta \right)=d\left(\hat{\theta }\right)+J\left(\hat{\theta }\right)\Delta \theta & +\frac{1}{2}\Delta \theta^{⊤}M\left(\hat{\theta }\right)\Delta \theta +O\left({∥\Delta \theta ∥}^{3}\right), \end{array}$$
where $J(\hat{\theta })$ and $M(\hat{\theta })$ are the Jacobian and Hessian evaluated at $\theta =\hat{\theta }$ and $\Delta \theta =\theta -\hat{\theta }$, respectively.

As a central feature of the ESM method, in Equation (Equation 11) the Hessian matrix must be expressed in terms of Jacobian. To enable this, the Jacobian is expanded in a first-order Taylor series and we obtain:(12)$$\begin{array}{cc} J(\theta ) & =J\left(\hat{\theta }\right)+\Delta \theta^{⊤}M\left(\hat{\theta }\right)+O\left({∥\Delta \theta ∥}^{2}\right)\Rightarrow \\ \Delta \theta^{⊤}M\left(\hat{\theta }\right) & =J\left(\theta \right)-J\left(\hat{\theta }\right)-O\left({∥\Delta \theta ∥}^{2}\right). \end{array}$$

Thus, by substituting $\Delta \theta^{⊤}M\left(\hat{\theta }\right)$ in Equation (Equation 11) with Equation (Equation 12) and dropping the second- and third-order residuals, we have:(13)$$d\left(\theta \right)≃d\left(\hat{\theta }\right)+\frac{1}{2}\left[J\left(\theta \right)+J\left(\hat{\theta }\right)\right]\Delta \theta .$$

Applying the optimization condition from Equation (Equation 8) and solving it, we find:(14)$$\begin{array}{cc} \nabla_{\Delta \theta }\left[\frac{1}{2}d{\left(\theta \right)}^{2}\right]{|}_{\Delta \theta =\Delta \theta^{\circ }} & =0\Rightarrow \\ \frac{1}{2}\left[J\left(\theta \right)+J\left(\hat{\theta }\right)\right]\Delta \theta^{\circ } & =-d(\hat{\theta }). \end{array}$$
where $\Delta \theta^{\circ }$ is the optimal increment that satisfies the optimization criterion.

The Jacobians of Equation (Equation 14) can be defined by applying the chain rule. Therefore, we have: (15)$$\begin{array}{cc} J(\hat{\theta }) & =J_{I}J_{w}J_{{\hat{n}}^{*}}, \end{array}$$(16)$$\begin{array}{cc} J(\theta ) & =J_{I^{*}}J_{w}J_{n^{*}}. \end{array}$$

Applying the same simplifications adopted by Silveira et al. [16], we assume that $J_{{\hat{n}}^{*}}\approx J_{n^{*}}$. Thus, Equation (Equation 14) becomes:(17)$$J_{\theta }\Delta \theta =-d\left(\hat{\theta }\right),$$
where $J_{\theta }=\frac{1}{2}\left(J_{I}+J_{I^{*}}\right)J_{w}J_{{\hat{n}}^{*}}$, $d(\hat{\theta })$ is the photometric error vector with all pixels applied in the optimization, $J_{I^{*}}$ and $J_{I}$ are the image gradient of the reference and current frames and $J_{w}$ is the matrix demonstrated by Benhimane and Malis [50]. $J_{{\hat{n}}^{*}}$ is the Jacobian of $H\left(n_{\rho }^{*}\left(\theta \right)\right)$ relative to the parameters of θ.

Finally, the increment of Δθ is given by:(18)$$\Delta \theta ={\left(J_{\theta }^{⊤}J_{\theta }\right)}^{-1}J_{\theta }\left(-d(\hat{\theta })\right).$$

#### 3.2.2. Plane Parameterization

There are several ways to parameterize a plane and the chosen parameterization influences the efficiency of the optimization. Following Lins et al. [48], the parameterization applied in this work uses the inverse depth information of three non-collinear pixels belonging to the reference image (Figure 2). Considering that these three points are in the same plane, the vector θ is defined as:(19)$$\theta ={\left[\begin{array}{ccc} \frac{1}{z_{1}^{*}} & \frac{1}{z_{2}^{*}} & \frac{1}{z_{3}^{*}} \end{array}\right]}^{⊤},$$
where $z_{1}^{*}$, $z_{2}^{*}$ and $z_{3}^{*}$ are the depth of these points.

For this parameterization, as introduced by Equation (Equation 9), the function which maps the parameters θ to $n_{\rho }^{*}$ is given by:(20)$$n_{\rho }^{*}=M\theta ,$$
where:(21)$$M=K^{⊤}{\left({\left[\begin{array}{ccc} p_{1}^{*} & p_{2}^{*} & p_{3}^{*} \end{array}\right]}^{⊤}\right)}^{-1},$$

$p_{1,2,3}^{*}$ are the pixel coordinates, in the reference image, of the three points and K is the matrix of intrinsic parameters. The update rule, given by f(·) function (Equation Equation 10) is:(22)$$\theta_{k+1}\leftarrow \theta_{k}+\Delta \theta .$$

#### 3.2.3. Pose Optimization

The problem of estimating the pose parameters is solved in the same way as plane optimization. In this case, Equation (Equation 7) becomes:(23)$$\min_{\zeta }=\frac{1}{2}\sum_{i=0}^{n-1}{\left\{I\left[w(H\left(T\left(\zeta \right)\right),p_{i}^{*})\right]-I^{*}\left[p_{i}^{*}\right]\right\}}^{2},$$
where T∈SE(3) represents the homogeneous transformation matrix belonging to the Lie group of rigid transformations. Similarly, the goal of this step is to find the unit dual quaternion $\zeta ={\left[q_{0},q_{1},...,q_{7}\right]}^{⊤}\in R^{8}$, which represents pose parameters and such that the reprojection error is minimal.

Using ESM, we find a similar result to that obtained in the plane optimization, given by:(24)$$J_{\mathrm{qzc}p}\Delta \zeta =-d\left(\hat{\zeta }\right),$$
where $J_{\mathrm{qzc}p}=\frac{1}{2}\left(J_{I}+J_{I^{*}}\right)J_{w}J_{\hat{T}}J_{\hat{\zeta }}$ represents the Jacobian of pose and the vector $d(\hat{\zeta })$ is the reprojection error. Finally, the pose increment is given by:(25)$$\Delta \zeta ={\left(J_{\mathrm{qzc}}p^{⊤}J_{\mathrm{qzc}p}\right)}^{-1}J_{\mathrm{qzc}p}\left(-d\left(\hat{\zeta }\right)\right).$$

#### 3.2.4. Unit Quaternion

Quaternions were introduced by Hamilton in 1866 as an extension of the complex number theory to formulate a four-dimensional manifold [38]. A quaternion is a four-component number consisting of a scalar part $q_{0}$ and a vector part $\bar{q}$. Formally, a general quaternion *q* is defined as:(26)$$\begin{array}{cc} q & =q_{0}+\bar{q}\\ \; & =q_{0}+q_{1}i+q_{2}j+q_{3}k, \end{array}$$
where its conjugate is given by $q^{*}=q_{0}-\bar{q}$. The vector form is $q=\left[q_{0}\;q_{1}\;q_{2}\;q_{3}\right]\in R^{4}$ and the orthogonal complex numbers $\vec{i}$, $\vec{j}$ and $\vec{k}$ are defined, such that ${\vec{i}}^{2}={\vec{j}}^{2}={\vec{k}}^{2}=\vec{i}\vec{j}\vec{k}=-1$.

The addition of two quaternions $a=a_{0}+\bar{a}$ and $b=b_{0}+\bar{b}$ is $a+b={\left[a_{0}+b_{0},\;a_{1}+b_{1},\;a_{2}+b_{2},\;a_{3}+b_{3}\right]}^{⊤}$ and the product of two quaternions is given by the Hamilton product, where:(27)$$ab=\left[\begin{array}{c} a_{0}b_{0}-a_{1}b_{1}-a_{2}b_{2}-a_{3}b_{3}\\ a_{0}b_{1}+a_{1}b_{0}+a_{2}b_{3}-a_{3}b_{2}\\ a_{0}b_{2}-a_{1}b_{3}+a_{2}b_{0}+a_{3}b_{1}\\ a_{0}b_{3}+a_{1}b_{2}-a_{2}b_{1}+a_{3}b_{0} \end{array}\right].$$

When the scalar part is equal to zero, the quaternion is written as $q=[0,\bar{q}]$ and is called a pure quaternion. Since $\bar{q}=q_{1}i+q_{2}j+q_{3}k$ is a three-dimensional vector, clearly there is a one-to-one correspondence between vectors in 3D space. In addition, a quaternion with a unit length, where ∥q∥=1, is called a unit quaternion and can be used to represent rotation about an axis (denoted by the unit vector n) and an angle θ as follows:(28)$$q=\cos\frac{\theta }{2}+n\sin\frac{\theta }{2}.$$

### 3.3. Unit Dual Quaternion

A dual quaternion, denoted by ζ, is defined as a dual number comprised of two quaternion components:(29)$$\begin{array}{cc} \zeta & =q_{r}+\epsilon q_{d}, \end{array}$$
where $q_{r}$ and $q_{d}$ are quaternions representing the real and dual parts, and ϵ denotes the dual unit, which is defined by $\epsilon^{2}=0$ and ϵ≠0. Equivalently, the dual quaternions can be defined as an eight-dimensional vector space:(30)$$\zeta ={\left[q_{0}\;q_{1}\;q_{2}\;q_{3}\;q_{4}\;q_{5}\;q_{6}\;q_{7}\right]}^{⊤},$$
where the first four elements represent the real part and the last four elements represent the dual part. The unit dual quaternion is subject to the constraint $\zeta \cdot \zeta^{*}=1$, where $\zeta^{*}=\left[{q_{r}}^{*}\;{q_{d}}^{*}\right]$ is its conjugate.

As the unit quaternions can be used to represent rotations, the unit dual quaternions can be used to represent rigid transformation [39]. Thus, the real part of the unit dual quaternion is a unit quaternion that represents rotation and the dual part represents translation combined with rotation:(31)$$q_{r}={\left[q_{0}\;\bar{q}\right]}^{⊤}$$
and
(32)$$q_{d}=\frac{1}{2}t\cdot q_{r},$$
where $t={\left[0\;\bar{t}\right]}^{⊤}$ is a pure quaternion formed with position vector.

To link the unit dual quaternions with optimization, the homogeneous transformation matrix T∈SE(3) is given in terms of the unit dual quaternion:(33)$$T\left(\zeta \right)=\left[\begin{array}{cc} R(q_{r}) & 2q_{d}\cdot {q_{r}}^{*}\\ 0 & 1 \end{array}\right]$$
and the update rule used in optimization is given by:(34)$$r_{k+1}\leftarrow r_{k}\cdot \Delta r=q_{r}\Delta q_{r}+\epsilon \left(q_{r}\Delta q_{d}+q_{d}\Delta q_{r}\right),$$
where the updated pose $r_{k+1}$ is obtained by the multiplication of the pose $r_{k}$ and the optimization increment Δr through the Hamilton product.

### 3.4. Summary of DSO

DSO is a keyframe-based approach, in which each keyframe is responsible for storing several high-gradient points, distributed over the image. Using a sliding window procedure, only a fixed amount of keyframes are kept in the system. Whenever a new keyframe needs to be created, an old keyframe is discarded. Each point of each keyframe has its inverse depth estimated through disparity, refined through Gauss–Newton optimization and kept in a semi-dense depth map.

A total of 2000 points, distributed along the keyframes belonging to the sliding window, are chosen and called active points. Whenever a new frame is available, these points are used to estimate the new pose. Later, the active points are used in the sliding window optimization, which is responsible for refining the pose estimates of all the keyframes of the sliding window and the depth parameters of the active points. In the global optimization, the intrinsic camera parameters and the affine brightness correction factors are also refined.

The energy function used in DSO is given by Equation (Equation 35):(35)$$E=\sum_{p^{*}\in N_{p}}\omega_{p}{∥\left(I\left[p\right]-b\right)-\frac{e^{a}}{e^{a^{*}}}\left(I^{*}\left[p^{*}\right]-b^{*}\right)∥}_{\gamma },$$
where p can be found by Equation (Equation 1), $a^{*}$, $b^{*}$, *a*, *b* are the brightness parameters of the reference and current images and $\omega_{p}$ is a gradient-dependent weighting factor. $N_{p}$ consists of the set including all pixels in the optimization. ${∥\cdot ∥}_{\gamma }$ represents the Huber norm.

As active points leave the scene or become occluded, they are marginalized and other points are activated. When many points of a single keyframe are discarded, the entire keyframe is marginalized and a new keyframe is inserted into the sliding window.

## 4. Direct Planar Odometry (DPO) with Stereo Camera

The proposed framework is based on DSO and is organized according to Figure 3. The system is divided into two parts: front-end and back-end. The front-end is composed of the system initialization, the tracking step and the creation and deletion of keyframes from the sliding window. The back-end is responsible for optimizing all the parameters belonging to the sliding window keyframes.

In the system startup, the first stereo pair is detected and a set of high-gradient pixels, distributed over the entire left image of the stereo pair, is selected. The inverse depth of each pixel is estimated through disparity and refined through optimization. Some of these points are used for planar region creation, which will be part of the first keyframe of the sliding window.

When the second stereo pair is available, the tracking algorithm estimates the relative pose between the keyframe and the current frame. Then, through a set of rules, the system checks if it is necessary to turn the current frame into a new keyframe. If so, points from the current frame are selected to form new immature planes, which are a set of planes from which the active planes are selected, i.e., the planes that will be part of the optimization. Moreover, whenever a new keyframe is created, the sliding window optimization is performed. Additionally, this step is also responsible for managing planar regions from a given keyframe inside the sliding window.

### 4.1. Initialization

The first step carried out in initialization is the selection and depth estimation of high-gradient points in the left image. The procedure for point and depth estimation is the same as in Stereo DSO [3]. These points are used to create planar regions in the image. This step is carried out by creating a set of triangles generated by the 2D Delaunay triangle algorithm. This routine differs from PVI-DSO [14], which uses 2D Delaunay triangulation over the DSO depth map.

Finally, the initialization is performed at different resolutions of the image (pyramid levels) and the estimated features are used in the tracking step. The planar regions generated at the original image resolution are activated and stored in the first frame, which is the first keyframe of the sliding window.

### 4.2. Tracking

As indicated in Figure 3, the tracking step is performed between the left camera images. Using the formulation presented in Equation (Equation 25), parameterized with unit dual quaternions (Section 3.3) and using planar regions, the relative pose between the current frame and the newest keyframe can be determined by the tracking algorithm. The active planes, applied in the tracking step of all keyframes in the sliding window, are projected in the newest keyframe. In addition, as in DSO, the constant motion model is used for estimating the initial pose and in optimization, an image pyramid approach is employed.

### 4.3. Keyframe Creation and Disposal

The creation and exclusion of new keyframes follow a similar rule to the one adopted in DSO [10]. Their work uses three criteria for creating a new keyframe: the optical flow mean squared error (MSE), the optical flow MSE without rotation and the relative brightness factor between two images. Since this work does not use brightness parameters, only the first two criteria are used to manage when new keyframes should be created.

In our approach, the management keyframe exclusion is effected with two criteria: the keyframe number of planar regions participating in the sliding window optimization and the keyframe age. Keyframes with fewer active planar regions and that are not among the two newest keyframes have priority to be discarded. Like the DSO, a limit of seven keyframes is defined to be kept in the sliding window.

### 4.4. Planar Region Creation and Disposal

In the DSO approach, when a new keyframe is inserted into the sliding window, a series of high-gradient points are selected from the left-image keyframe. Some of these selected points are observations of keyframe points, inside the sliding window. To prevent these points from being inserted in the newest keyframe, the DSO uses a *distance map*. It expresses the distances of map elements to points of all sliding window keyframes, projected onto the image of the newest keyframe. Thus, the system can discard points with the same image position as existing points.

In contrast, in the construction of the DPO map, only active planes are used, i.e., planar regions of all keyframes considered in the sliding window to participate in the optimization. As they are composed of three points, the vertices of these regions are projected onto the image of the newest keyframe with halved resolution. This resolution reduction procedure is equivalent to expanding the vertex size in the newest image. The position in the image of each projected vertex is used in the distance map, which has the same dimensions as the reduced image.

All elements of the distance map are initialized with the integer 1000, representing that still there are no nearby neighbor projected planes. Then, the indexes of the planar region vertices are used to attribute zero in the corresponding distance map elements, a value that indicates occupied positions. A rasterization process is also performed on each planar region to insert zero in the inner parts of these planes, indicating that these elements are also occupied positions.

To attribute values for other map elements, a point is chosen for each planar region edge. These points are defined by the intersection of the edge with the line formed by the region centroid and the remaining vertex, which is not part of the edge. From these points, elements neighboring an element that has a smaller integer number get the smaller number added to the number 1. Thus, this procedure allows the distance map to report the distance from an element of the array to an edge of a planar region.

Each time a new keyframe is created, high-gradient points distributed throughout the left image are selected. The depth of these points is estimated through disparity. Two-dimensional Delaunay triangularization is applied to these points to generate a set of triangles, which are considered immature planes whenever the depths of their vertices are valid.

Similar to DSO, part of these planes can be activated and included in the optimization, according to the maximum number of regions. To be activated, an immature plane cannot be an observation of another active plane. To achieve this, before a plan is activated, a check of its position with respect to other active points is performed using the distance map. Figure 4 shows an example of how new planar regions are created.

When the number of active planes is greater than the defined maximum number, the ones that leave the image or have a high residual are discarded. This procedure differs from the one performed with DSO. In their method, parameters from the keyframe that will be discarded are used one last time in the sliding window optimization.

### 4.5. Sliding Window Optimization

The optimization is performed with all active planes of all keyframes in the sliding window. When a given plane of an active keyframe is visible in another one, we consider this an observation. Thus, in this approach, an observation always takes place between two keyframes. In addition, since the sliding window keyframes have geometric proximity, a given planar region can have multiple observations.

The formulation of the sliding window optimization can be represented by:(36)$$\min_{\Phi }=\frac{1}{2}\sum_{k\in F}\sum_{\theta_{i}\in \Theta }\sum_{j\in \mathrm{obs}(\theta_{i})}d{\left(\zeta_{jk},\theta_{i}\right)}^{2},$$
where $\Phi ={\left[\Phi_{\mathrm{qzc}}p,\;\Phi_{\theta }\right]}^{⊤}$ is the vector with all optimized parameters, $\Phi_{\mathrm{qzc}}p$ is the vector of pose parameters, $\Phi_{\theta }$ is the vector of feature parameters, F represents the set of keyframes in the sliding window, Θ is the set of active planar regions of a keyframe and $\mathrm{obs}(\theta_{i})$ is the number of observations of a given planar region $\theta_{i}$.

Since the pose parameters of the stereo pair are fixed, the feature observations realized by the stereo pair are not directly added to the optimization. To take into account the information of these observations, we perform a similar procedure carried out with Stereo DSO. This procedure consists of replacing the residual d of the temporal stereo observations with:(37)$$d{\left(\zeta_{jk},\theta_{i}\right)}^{\prime }=\lambda d\left(\zeta_{RL},\theta_{i}\right)+\left(1-\lambda \right)d\left(\zeta_{jk},\theta_{i}\right),$$
where $d{\left(\zeta_{jk},\theta_{i}\right)}^{\prime }$ is the weighted average of the static and temporal residuals, given that λ is a constant fixed in 0.15 and $\zeta_{jk}$ represents the relative pose between the keyframe *k* and the *j*-th observation.

The increment ΔΦ of the parameters is given by:(38)$$\Delta \Phi ={\left[\Delta \zeta ,\;\Delta \theta \right]}^{⊤}={\left(J^{⊤}J\right)}^{-1}J\left(-d^{\prime }\right),$$
where $J={\left[J_{\mathrm{qzc}}p,J_{\theta }\right]}^{⊤}$ represents the Jacobian matrices of the relative poses and active planar regions and $d^{\prime }={\left[d_{1}^{\prime },d_{2}^{\prime },...,d_{n}^{\prime }\right]}^{⊤}$ represents the residual vector of all observations.

As the DSO, this formulation also uses the concept of First Estimate Jacobian [51] in the assembly of the Hessian matrix and Shur’s elimination; these are done similarly. In addition, in their method, the marginalized variables in Shur’s elimination are points that will be discarded. The proposed technique, on the other hand, marginalizes the active features in the same manner as is common in indirect methods [52]. Moreover, since the terms of the residuals in the optimization matrix depend on two frames, the adjoint dual unit quaternion is used to relate relative pose to absolute pose [53]. An example of a configuration of the sliding window can be viewed in Figure 5.

## 5. Experimental Results

For performance evaluation and comparison of the method, six sequences from the KITTI dataset [54] were used. In each of these sequences, an average trajectory was used, defined by the average of ten simulations. As an evaluation criterion, the Absolute Pose Error (APE), implemented in Python’s EVO library [55], was used. This metric compares the reference poses (ground truth) with the poses estimated with the VO/SLAM method. For the simulations, whenever possible, we tried to use the same configuration parameters as the DSO in the DPO. Thus, parameters such as the number of pixels per point or vertex were set to eight (8) and the maximum number of active keyframes within the sliding window was seven (7). The results for each sequence can be viewed in Table 1, Table 2 and Table 3. For each sequence, the APE metric is given for each frame of the sequence. Information such as RMS error, mean, median and standard deviation are also presented.

### Discussion

Analyzing the graphs of sequences 3, 4, 6, 7, 9 and 10 from the KITTI dataset, it is possible to notice in Table 4 that in sequences 3, 4 and 7 the mean errors and standard deviations were smaller in the DPO. In sequences 6 and 9, DPO mean errors were slightly above the DSO. However, the maximum errors of the DPO were much larger than the DSO, indicating large errors at isolated points of the trajectory. Finally, sequence 10 showed a much smaller mean error and standard deviation for DSO. It is important to emphasize that even without photometric camera calibration and optimization of brightness parameters, the DPO presented a similar performance to the DSO and obtained better results in half of the evaluated cases. However, of the six analyzed sequences, three presented a much higher maximum error for DPO, a fact that might be partially justified by the lack of optimization of illumination parameters. In the future, we intend to insert an illumination optimization model into the proposed technique.

From a computational cost point of view, we measure the frame per second (FPS) for both methods. Although DPO was programmed on the stereo DSO platform, no multi-threading parallelization or SSE (Streaming SIMD Extensions) instruction set was used to accelerate the algorithm. Nevertheless, on a 7th-generation Intel core-i5 PC with 8 GB RAM, the algorithm runs at a rate of about 1 FPS, while DSO runs at a rate of 5.63 FPS. Looking at the computational cost from the perspective of the amount of data used in the optimization, the number of active planes is limited to 667 planes in the sliding window. Since each plane consists of three points, the maximum number of points per optimization is 2001 points, equivalent to the number used by Stereo DSO.

Figure 6 presents an example of the map reconstructed by our method. From a set of points, the planes used in the optimization are estimated from the Delaunay triangularization. Although this approach is ideal for coupling vertices that belong to different planes, for the sake of simplicity, in this first version the generated planes were considered independent. From the map point of view, this work is seen as a first step in the development of a map that can be used not only for odometry but also to represent structures in the environment to aid trajectory planning. Although some of the planes determined by the method may not represent real-world planes, Delaunay triangularization allows non-planar regions to be approximated by planes.

## 6. Conclusions

This paper presented a technique inspired by DSO, but instead of points, we make use of planar regions as features. It also introduced the use of dual quaternions to represent the camera’s poses in direct methods. Comparisons of the proposed technique with Stereo DSO were performed and the presented results showed DPO accuracy within the DSO standards, this being consistently superior in half of the experiments performed, given that each sequence was evaluated several times. In the future, in order to improve the accuracy of the odometry, we intend to add illumination parameters to the optimization, as well as loop closure detection. Moreover, in order to improve the map representation, we intend to make the features more susceptible to the inclusion of semantics, which could help in the trajectory generation.

## Figures and Tables

**Figure 1 sensors-23-01393-f001:**
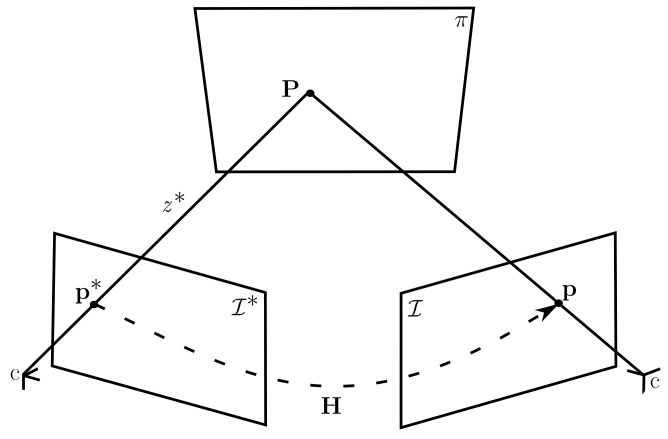
Homography between two images given a plane.

**Figure 2 sensors-23-01393-f002:**
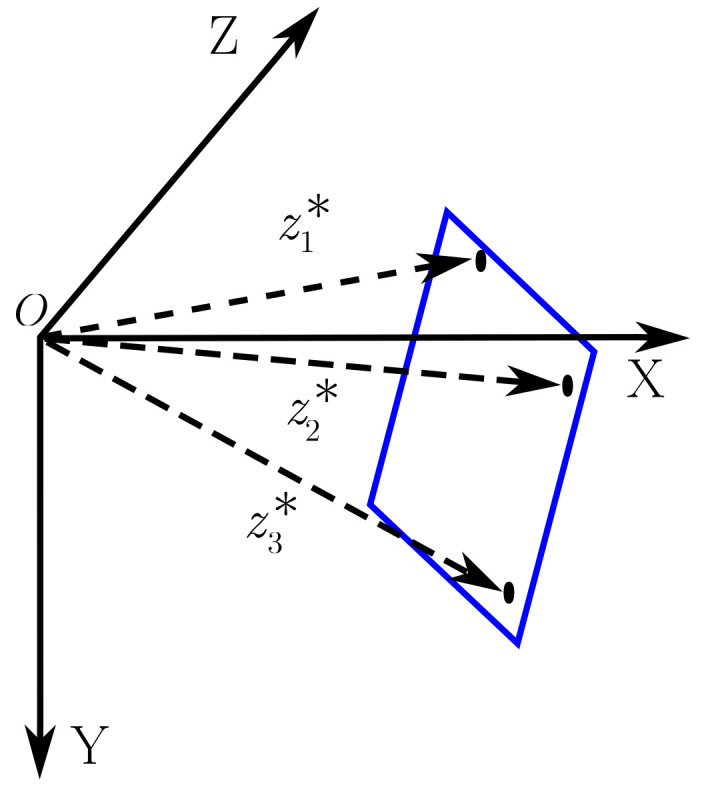
The figure shows three non-collinear points, belonging to the plane π and projected onto the reference image $I^{*}$ with depths $z_{1}^{*}$, $z_{2}^{*}$ and $z_{3}^{*}$, respectively.

**Figure 3 sensors-23-01393-f003:**
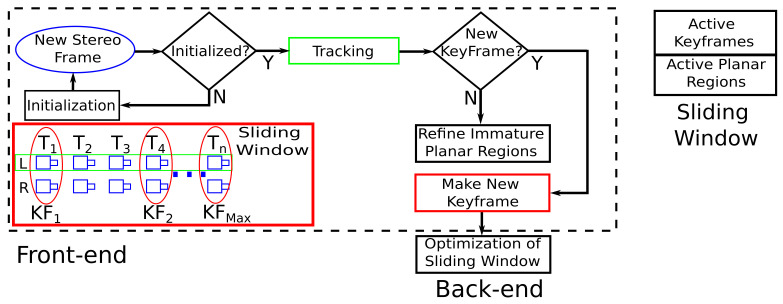
Framework of the proposed approach.

**Figure 4 sensors-23-01393-f004:**
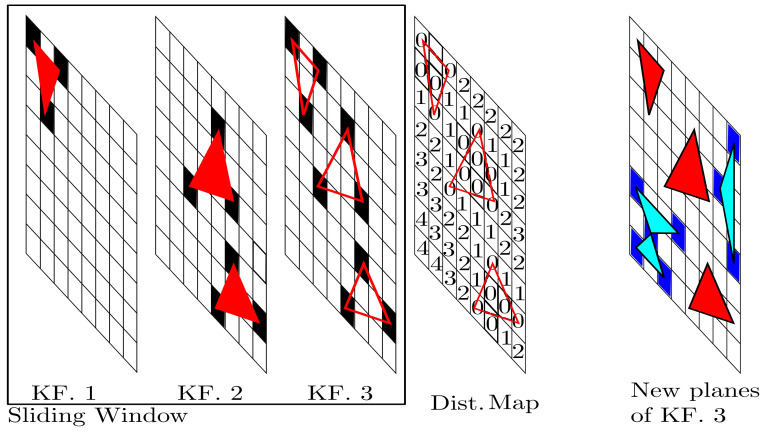
The example shows a sliding window formed by two old keyframes and one new keyframe. In the first keyframe (KF. 1), there is an active planar region, represented by the triangle (red) and its vertices (black). The second keyframe (KF. 2) has two active planar regions. The third keyframe (KF. 3) shows projections of the vertices from the previous keyframes (black). The distance map is exemplified in the fourth frame. Elements with zero indicate occupied positions with old edges. Elements with values above zero represent the distance from the element to the edge. Thus, new planes that fall at a certain distance from the edges of old activated planes are accepted to be activated in the new keyframe. Finally, the last frame presents new activated planes (cyan) and their vertices (blue).

**Figure 5 sensors-23-01393-f005:**
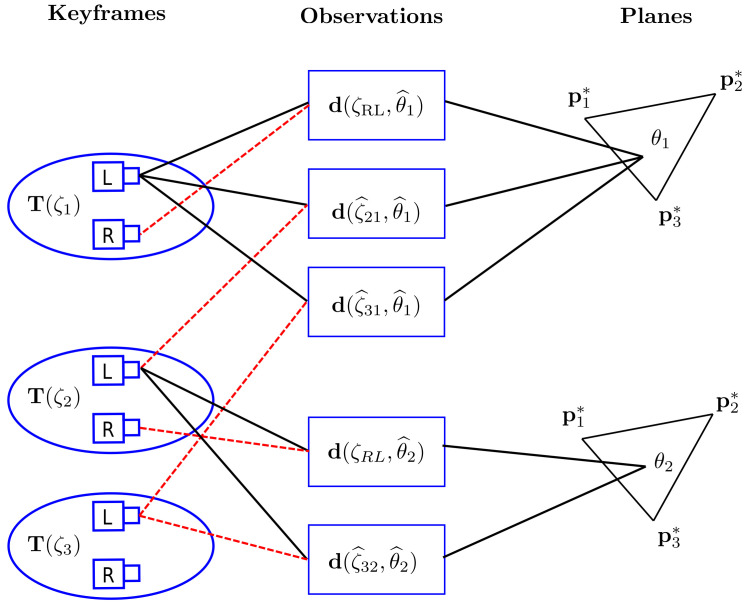
The example shows a sliding window with three keyframes, with poses $T(\zeta_{1})$,$T(\zeta_{2})$ and $T(\zeta_{3})$. There are two planar regions, $\theta_{1}$ belonging to the first keyframe, having three observations, and $\theta_{2}$ belonging to the second keyframe, having two observations.

**Figure 6 sensors-23-01393-f006:**
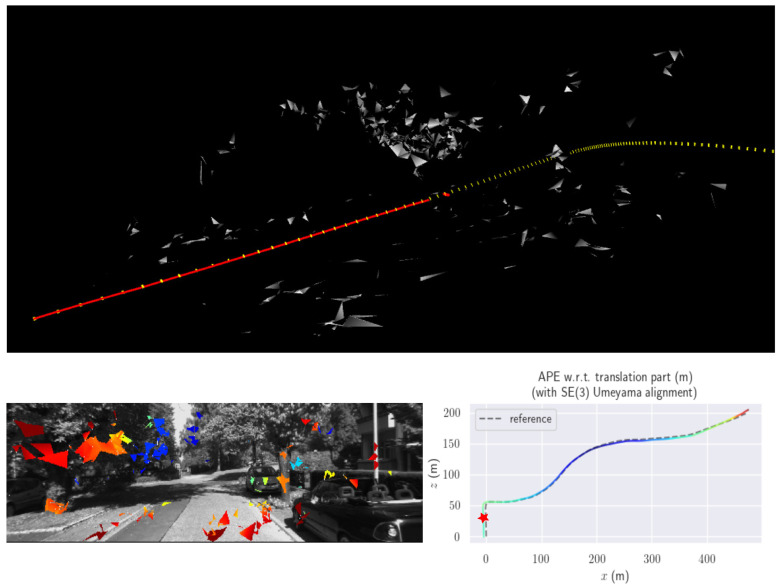
This figure shows a sample result, taken from KITTI sequence 3. The bottom right image represents the resulting DPO trajectory. Sections with colors closer to blue indicate less positioning error. The star indicates the position of the keyframe, shown in the lower left image. Finally, the top image presents a sample 3D visualization of the map.

**Table 1 sensors-23-01393-t001:** The absolute pose error (APE) for all frames of sequences 3 and 4 of the KITTI dataset.

DPO	Stereo DSO
Sequence 3	
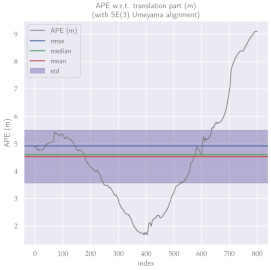	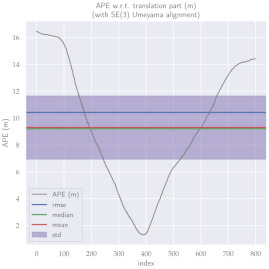
Sequence 4	
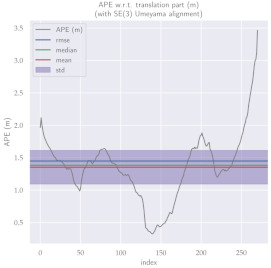	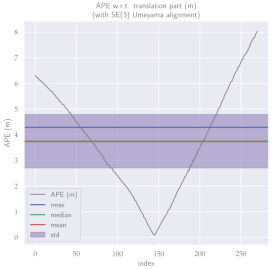

**Table 2 sensors-23-01393-t002:** The absolute pose error (APE) for all frames of sequences 6 and 7 of the KITTI dataset.

DPO	Stereo DSO
Sequence 6	
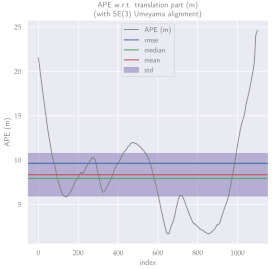	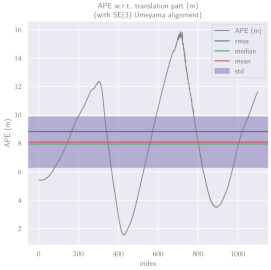
Sequence 7	
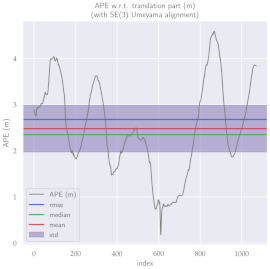	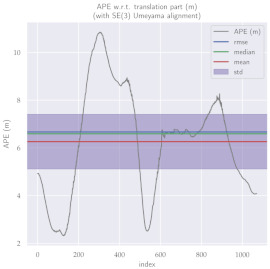

**Table 3 sensors-23-01393-t003:** The absolute pose error (APE) for all frames of sequences 9 and 10 of the KITTI dataset.

DPO	Stereo DSO
Sequence 9	
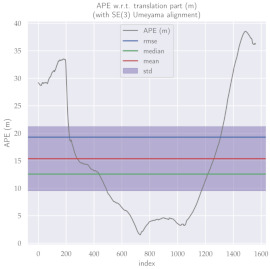	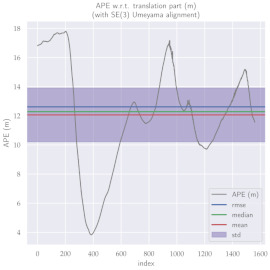
Sequence 10	
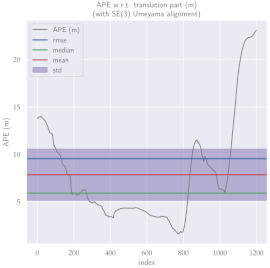	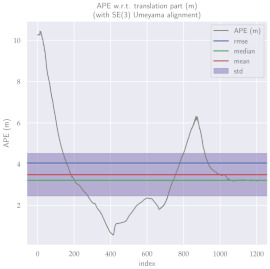

**Table 4 sensors-23-01393-t004:** Comparison of mean error, standard deviation and minimum and maximum error of the six analyzed sequences between DPO and Stereo DSO.

Absolute Pose Error (APE)						
**Seq.**	**Mean ± Std.**		**Min.**		**Max.**	
	**DPO**	**DSO**	**DPO**	**DSO**	**DPO**	**DSO**
3	4.53±1.92	9.29±4.71	1.68	1.32	9.10	2.46
4	1.35±0.51	3.74±2.08	0.32	0.09	3.46	8.03
6	8.35±4.80	8.08±3.55	1.69	1.55	24.61	15.87
7	2.48±1.01	6.26±2.29	0.18	2.31	4.59	10.84
9	15.37±11.64	12.05±3.70	1.52	3.82	38.54	17.80
10	7.85±5.42	3.48±2.06	1.61	0.56	23.07	10.43

## Data Availability

Not applicable.

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
