# Peer review of "DPO: Direct Planar Odometry with Stereo Camera"

_sensors, 2023, doi:10.3390/s23031393_

Round 1
Reviewer 1 Report
1. The article seems a bit lengthy. Please try to simplify the introduction and related works. The work quoted should be representative and correspond to the content of the article. It is unnecessary to introduce some basic works too much.
2. Please note that the font size in the figures should be consistent, such as Figs. 1, 2, 5, and the pictures in Table 1, which would cause inconvenience to the readers.
3. Could you supplement the comparison experiments about running speed and memory occupation between the proposed method and the comparison method? For engineers, they may be more concerned about them.
Reviewer 2 Report
1-The introduction with the related works is too long.
2- The author mentioned that the use optimization technique. Which type of optimization is used?
3- The authors mentioned that the presented work is compared with previous ones, I didn't mention any comparison.
4-The organization of the paper need to be reconsidered.
Reviewer 3 Report
Reviewers’ Comments
Manuscript ID: sensors-2098131
Title: DPO: Direct Planar Odometry with Stereo Camera
Authors: Filipe C. A. Lins * , Nícolas S. Rosa , Valdir Grassi , Adelardo A. D. Medeiros , Pablo Javier Alsina
Comments:
Visual odometry is useful for robot navigation in various industrial applications. This paper presents a direct planar odometry with stereo camera. However, the accuracy of the stereo vision technique is affected by the measurement distance, the illumination variations, and the extrinsic parameters of two cameras. The influences of those factors need to be investigated. And the advantage of the proposed method needs to be clearly stated in the paper. This paper needs to be further improved by following comments.
Specific comments:
(1) The main content of this paper is to develop direct planar odometry for robot navigation applications with stereo vision techniques. The effect of weather conditions (i.e., illumination changes, foggy day, temperature) and the camera motion itself need to be further investigated.
(2) The effectiveness of the developed methodology has been verified by one experiment, extensive experiments need to be added to demonstrate the robustness of the method.
(3) In field applications, the moving vehicles, pedestrians, and buildings will obstruct the camera view, how to solve this problem?
(4) The computational efficiency of the developed method also needs to be compared with conventional method.
Round 2
Reviewer 1 Report
I have no further suggestions
Reviewer 2 Report
The authors have addressed all the comments.